# Colostrum Therapy for Human Gastrointestinal Health and Disease

**DOI:** 10.3390/nu13061956

**Published:** 2021-06-07

**Authors:** Kanta Chandwe, Paul Kelly

**Affiliations:** 1Tropical Gastroenterology & Nutrition Group, University of Zambia School of Medicine, Lusaka 10101, Zambia; kantachandwe@gmail.com; 2Blizard Institute, Barts & The London School of Medicine, Queen Mary University of London, London E1 2AT, UK

**Keywords:** colostrum, diarrhea, inflammatory bowel disease, enteropathy, mucosal healing

## Abstract

There is increasing awareness that a broad range of gastrointestinal diseases, and some systemic diseases, are characterized by failure of the mucosal barrier. Bovine colostrum is a complex biological fluid replete with growth factors, nutrients, hormones, and paracrine factors which have a range of properties likely to contribute to mucosal healing in a wide range of infective, inflammatory, and injury conditions. In this review, we describe the anatomy and physiology of the intestinal barrier and how it may fail. We survey selected diseases in which disordered barrier function contributes to disease pathogenesis or progression, and review the evidence for or against efficacy of bovine colostrum in management. These disorders include enteropathy due to non-steroidal anti-inflammatory drugs (NSAIDs), inflammatory bowel disease (IBD), necrotizing enterocolitis, infectious diarrhea, intestinal failure, and damage due to cancer therapy. In animal models, bovine colostrum benefits NSAID enteropathy, IBD, and intestinal failure. In human trials, there is substantial evidence of efficacy of bovine colostrum in inflammatory bowel disease and in infectious diarrhea. Given the robust scientific rationale for using bovine colostrum as a promoter of mucosal healing, further work is needed to define its role in therapy.

## 1. Introduction

Gastrointestinal and hepatobiliary diseases rank high among the contributors to burden of disease in industrialized countries. Major costs are associated with hepatitis, esophageal disease, abdominal pain, and inflammatory bowel disease [1,2]. Gastrointestinal cancers [3] and inflammatory bowel disease (IBD) [4,5] appear to be increasing in low- and middle-income countries too, though under-diagnosis introduces uncertainty into available estimates.

Many gastrointestinal diseases are characterized by loss of mucosal integrity. This includes macroscopic ulceration in peptic ulceration, IBD, reflux esophagitis, cancers, caustic injury, esophageal candidiasis, and tuberculosis. In some disorders (coeliac disease and other enteropathies, radiation- or chemotherapy-related mucositis, bowel ischemia), the loss of integrity may be visible only on a microscopic scale or may require functional testing (for example, permeability testing using lactulose) to detect [6,7]. Intestinal barrier failure detectable in IBD [8], enteropathies [9], critical care [10], and mucositis [11] leads to microbial translocation of bacteria, viruses, and their pathogen-associated molecular pattern molecules (PAMPs) from gut lumen to systemic circulation and the lymphatic system [12,13]. Microbial translocation then drives systemic inflammation [6,12,13]. There is also increasing awareness of the adverse consequences of mucosal barrier failure in disorders not primarily originating in the gut, notably sepsis in critical illness [10] and HIV infection [12,13]. Furthermore, several disorders of the liver are accelerated by microbial translocation [14], including alcoholic [15] and non-alcoholic fatty liver disease (NAFLD) [16].

This narrative review sets out to explore the landscape for using bovine colostrum (BC) as therapy for GI disorders, particularly those characterized by mucosal barrier failure. We therefore begin by reviewing the mucosal barrier and its repair mechanisms, which provide a rationale for the use of BC as therapy. Composition of BC has been comprehensively reviewed in a companion paper in this supplement [17]. Hyperimmune Bovine Colostrum (HBC) or extracted immunoglobulins for treatment of intestinal infection have been used successfully for treatment of several intestinal infections including rotavirus, *Cryptosporidium* spp., *Shigella* spp., and *Clostridium difficile* [18,19], but these will not be the focus of this article. We searched for both human studies and those using animal models in PubMed, using key words including “colostrum” and “diarrhea”, (“IBD” or “Crohn’s disease” or “ulcerative colitis”), “necrotizing enterocolitis”, “short bowel syndrome”, “gut inflammation”, “NSAID-induced gut injury”, (“cancer” or “malignancy”).

## 2. The Mucosal Barrier in the Gut

The primary function of the gut is digestion and absorption of nutrients from food. However, the gut is also an ecological niche for billions of bacteria, viruses, protozoa, and fungi: the microbiota. Ingested food also contains bacteria, viruses, protozoa, and fungi, some of which may be pathogenic. The mucosal barrier refers to the anatomical and functional boundary between the host and the environment enclosed within the gut lumen (Figure 1). It comprises several elements:The gastric acid barrier limiting access to the intestine;The mucus layer [20], which contains a diffusion gradient of antimicrobial factors (nitric oxide, defensins, other antimicrobial peptides [21]) and immunoglobulins (mainly secretory IgA);The epithelium, a single cell monolayer joined by tight junctions and other cell adhesion structures [22], and which is continuously replaced from the crypts;Intra-epithelial lymphocytes (IELs), a group of resident lymphocytes;Lymphocytes and macrophages in the lamina propria, which circulate to mesenteric lymph nodes and provide anamnestic immunity;Downstream, the macrophage (Kupffer cell) compartment of the liver provides a barrier against pathogens and pathogen-associated molecular patterns (PAMPs), which escape the first four elements of the barrier listed above. This will not be discussed further;The microbiota, which confers resistance to extraneous colonization.

Nevertheless, pathogens and toxins do succeed in breaching the mucosal barrier, frequently when exposure is high, and the mucosa of the gastrointestinal tract has evolved a range of mechanisms to prevent serious consequences [20].

## 3. Gastrointestinal Repair Mechanisms

The epithelium of the gut has one of the fastest turnover rates of all cell compartments of the human body. Thus, with a turnover time of 3–5 days, cells which have been damaged by pathogens or toxins are rapidly replaced. This critically important feature of intestinal physiology allows complete recovery from life-threatening toxic attack during, for example, cholera, when a substantial fraction of small intestinal enterocytes may have been irreversibly intoxicated and committed to chloride secretion. Their replacement leads to clinical recovery. The high turnover of the intestinal epithelium is dependent on luminal nutrition, and starvation (withdrawal of luminal nutrients) leads to atrophy and impairment of barrier function [23]. This is at least partly under the control of glucagon-like peptide 2 (GLP2), a product of the glucagon gene, which has important trophic effects on the intestine and therapeutic administration of which can drive mucosal hypertrophy even when luminal nutrients have been withdrawn [24]. The actions of GLP2 are indirect, mediated largely through neural and myofibroblast secretion of insulin-like growth factor-1 (IGF-1) [25].

Ulceration is a discontinuity in the epithelial surface. The factors which permit ulcers to persist are unknown, but healing (restitution) of wounds in epithelial monolayers in vitro has been well studied. Once a monolayer is wounded, cell migration from the margins, accompanied by increased cell turnover, permits the breach to be covered and integrity to be restored [26]. In vivo, injury is associated by dedifferentiation and emergence of injury-associated cell types, including surface mucosal cells, ulcer-associated cells, mucosal neck cells, spasmolytic polypeptide-expressing metaplasia (SPEM), and pyloric metaplasia [27]. Intriguingly, many of these injury-associated metaplastic cells generate mucins, which suggests that restoration of the mucus layer is a key process in restitution. Polyamines drive epithelial restitution through Ca^2+^ signaling [28,29]. Peptide growth factors are also required for these processes [17], including hormones (LnRH), cytokines, IGF-1, EGF, TGFα and β, PDGF, GH and GHRF, as well as milk fat globule proteins [30]. Colostrum provides many of these factors and enhances epithelial restitution [31]. This provides a clear rationale for exploring the potential therapeutic use of colostrum for disorders characterized by loss of epithelial integrity. It was observed many years ago that expression of receptors for many growth factors is restricted to the basolateral surface of the enterocyte, suggesting that trophic factors only act on the epithelium when its integrity has been breached; so-called luminal surveillance [32]. Many of these factors have anti-inflammatory properties [33]. Innate immune receptors now also appear to play a role in restitution and repair [34]. While EGF appeared to be the key driver of epithelial repair, there is now evidence that other peptides, such as Neuregulin-1, which signal through EGFR-related pathways, may in fact be more potent [35].

## 4. Clinical Applications: Colostrum as Therapy

Bovine colostrum has been evaluated for efficacy in a range of clinical disorders of the stomach and intestine. One caveat which must be introduced prior to exploring the evidence is that not all colostrum preparations are equally effective when tested by bioassay (as opposed to merely measuring immunoreactivity of component peptides and proteins), which may explain some variation in clinical study results (Table 1) [36].

### 4.1. NSAID-Induced Gastrointestinal Injury

In an experimental model of NSAID-induced mucosal damage, BC reduced gastric injury (assessed histologically) by up to 60% [37]. This effect could be replicated using TGFβ in similar amounts to that calculated to be present in the BC, and colostrum also attenuated the reduction in small intestinal villus height due to indomethacin. Similar results were obtained in a diclofenac model of intestinal injury in the rat, and a synergistic effect of glutamine was also noted [38]. In an indomethacin-induced injury model in mice, villus height was also increased by the use of bovine colostrum (in this experiment collected only after 5 days post-partum) [39]. Evidence for a similar protective effect in humans is slender. In a small study in human volunteers, BC prevented the rise in permeability induced by indomethacin treatment [40]. However, in long-term use, the rise in permeability is barely measurable, and in the same study [40], no effect was observed in this context. The authors concluded that the small intestine may undergo some degree of adaptation, but there is as yet no direct evidence for this, and long-term NSAID users experience ongoing blood loss. Further studies are needed of the effect of colostrum (with or without other constituents) on NSAID enteropathy, as new approaches are needed to this clinical problem.

### 4.2. Inflammatory Bowel Disease (IBD)

Inflammatory bowel disease (IBD) is a chronic, relapsing inflammatory disease of the gastrointestinal tract that affects both children and adults [72,73,74]. Two major types are described, ulcerative colitis and Crohn’s disease, but minor forms (collagenous colitis, lymphocytic colitis) are also important. The cause of IBD is not known but host genetic factors, microbiota, and environmental factors are believed to interact, leading to an adverse immune response in the GI tract. Treatment options for IBD include aminosalicylates, antibiotics, steroids, immunomodulators, stem cell therapies, and surgery [75,76,77]. However, none of these treatments are effective in all patients.

Bovine colostrum has shown some promising results in reducing inflammation and symptoms in both animals and humans. In murine models of colitis, BC and its components have been shown to prevent or reduce chemically induced colitis [39,41,42,43,44,45]. In a study by Khan et al., BC was shown to improve symptoms and histological scores of patients with distal colitis who received colostrum enemas in addition to mesalazine, compared to controls who only received mesalazine [46]. Benefits have also been observed in children with Crohn’s disease who receive nutritional supplements rich in TGF-β. Such formulas are not only associated with improvement in the pediatric Crohn’s disease activity index (PCDAI), but also in BMI and inflammatory markers [78,79]. TGF-β is a one of the major growth factors present in colostrum and such results may point to the potential benefit of BC as an adjunct therapy in patients with IBD.

### 4.3. Infectious Diarrhea

Diarrhea is a common problem in both adults and children. In immunocompetent and well-nourished adults and children, diarrhea is usually self-limiting, but in malnourished children and immunocompromised adults, it is more likely to persist, leading to worsening malnutrition with metabolic and immune consequences. After the neonatal period, diarrhea is the second leading cause of death for those under the age of 5 years [80]. Diarrhea also predisposes children to undernutrition, which may have long-term consequences, such as poor neurocognitive development [81,82]. Treatment of diarrhea focuses on maintaining fluid and electrolyte balance as well as nutrition. Breastfeeding is well recognized to protect against diarrhea incidence, hospitalization, and mortality [83,84].

The high concentration of immunoglobulins and antimicrobial factors present in colostrum compared to mature milk may be of additional benefit in prevention and treatment of infectious diarrhea in both children and adults, especially the immunocompromised or malnourished. There is evidence for in vitro efficacy from two studies using cultured Caco-2 cells which showed that BC, with or without egg, can reduce the pathogenic effects of a range of pathogens [47,85].

A meta-analysis of 5 randomized control trials with a total of 324 children was conducted by Li et al. [48]. The pooled analysis included healthy children, those with *rotavirus* and *Escherichia coli* diarrhea and hospitalized children without diarrhea [48]. BC treatment was associated with reductions in stool frequency and occurrence of diarrhea compared to placebo. Since that analysis was undertaken, Barakat et al. [49] evaluated the effect of giving BC to children under 5 years in Egypt who had acute diarrhea. In this double-blind randomized controlled trial, 160 children were enrolled with half receiving BC in addition to standard care for acute diarrhea, while the other half only received standard care. The BC group showed reduced frequency of vomiting, diarrhea, and reduced Vesikari (clinical severity) scores after 48 h as compared to the control group; after 1 week, none of the children in the BC group had diarrhea compared to the control group, where 13% of the children still had diarrhea (*p* = 0.001). In Guatemala, duration of diarrhea was measured in 301 children randomized to either a novel treatment containing specific antimicrobial factors derived from BC and hen’s egg or placebo [50]. Although there was no effect on the primary endpoint, in a subset of children with the targeted organisms, the treatment was able to significantly shorten the diarrhea duration compared to placebo. In another study from Egypt that included 160 children aged 1–6 years, Saad et al. observed a reduction in diarrhea and URTI episodes in children receiving BC [51]. Such results are very encouraging because the children who experience the worst diarrhea outcomes usually have recurrent or persistent diarrhea and mainly reside in developing countries. The use of BC would be acceptable in most cultures in addition to being relatively cheap and safe. A number of other studies have looked at the use of BC in children with diarrhea. Even though some studies have shown little or no effect of BC on diarrhea frequency and duration, and some of the beneficial effects reported were not the primary endpoints of the trials, the majority of the studies have shown that BC has beneficial effects in diarrhea treatment [86]. Of the 18 studies reported, none showed worse outcomes in patients treated with BC, suggesting a good safety record [86].

In adults, a randomized control trial involving 90 healthy adult volunteers showed that BC is able to prevent diarrhea caused by enterotoxigenic *Escherichia coli* (ETEC) [52]. This study had two parts, the first of which comprised of 30 volunteers aged between 18 and 40 years. Fifteen (15) of these volunteers received BC tablets three times a day while the other 15 received placebos. On the third day, all the participants were given a solution containing ~10^9^ colony forming units of ETEC. Only 1 (7%) volunteer in the BC group developed diarrhea compared to 11 (73%) in the placebo group (*p* = 0.005). Components in BC were postulated to have interfered with the binding of ETEC to epithelial cells. In a RCT in adult patients with HIV-related diarrhea in Uganda, BC reduced stool frequency and fatigue, and permitted weight gain compared to routine care alone [53]. In a small (n = 62) RCT in Iran in patients in an ICU setting, patients randomized to BC had reduced microbial translocation (measured by LPS and zonulin) and experienced less diarrhea than patients randomized to placebo [54].

Another condition characterized by polymicrobial intestinal infections, environmental enteropathy, was the subject of another randomized controlled trial of BC with or without egg in Malawi [55].

### 4.4. Short Bowel Syndrome; Intestinal Failure

Short bowel syndrome (SBS) is a debilitating malabsorption disorder arising from a reduced functional small intestine. It usually arises from resection of large segments of the small bowel secondary to conditions such as necrotizing enterocolitis (NEC), midgut volvulus, mesenteric ischemia, trauma, or IBD [87,88,89]. Patients present with malabsorption and malnutrition with prognosis largely depending on the amount of functional bowel present, anatomy, and the ability of the remaining bowel to adapt. Currently, there is no cure for SBS, and treatment options are costly in addition to adverse side effects in the long term. Treatment options include total parenteral nutrition, modified diets, and drugs that enhance intestinal adaption (growth hormone, teduglutide, and glutamine) [90,91].

The growth factors and other bioactive molecules in BC make it an attractive option as a treatment adjuvant for promoting intestinal adaptation in SBS treatment. Promising results from animal models of SBS suggest that the supplementation of diets with BC led to weight gain and signs of intestinal adaptation [56,57,58,59]. However, results of clinical trials have so far not shown similar results in terms of improvements in intestinal function in children and adults [60,61,92], perhaps because measurement of endpoints requiring invasive sampling is more difficult than in animal models. Further clinical trials are needed to identify the dosage or component/s of BC that may benefit patients with SBS.

### 4.5. Necrotizing Enterocolitis

Necrotizing enterocolitis (NEC) is a devastating intestinal disease that mainly affects premature infants [62,93]. NEC is associated with mortality rates of 20–30%, with those requiring surgery having the highest rates. The etiology is most likely multifactorial with gestational age, reperfusion injury, and a proinflammation response playing a major role. Treatment outcomes are usually poor and so there has been a focus on prevention and early detection of NEC. The antibacterial, anti-inflammatory, and growth factors present in BC has made it an attractive candidate for the prevention of NEC with promising results noted in animal models [63,64,65]. Human studies have so far not shown a significant benefit of using colostrum or its components in reducing the incidence of, or mortality from, NEC in preterm infants [66,67,68,69,70,71]. These conclusions are consistent with a recent meta-analysis [94], but the meta-analysis did show a promising reduction in the time required to reach full enteral feeding, an important stage in clinical recovery [94]. Clearly, future studies are needed to define indications for BC in combination with other factors.

### 4.6. Intestinal Consequences of Cancer Treatment

In a pig model of doxorubicin-induced mucositis, colostrum reduced histological lesions and improved the damage to disaccharidase and glucose uptake [95]. An RCT of bovine colostrum in children undergoing chemotherapy for acute lymphocytic leukemia showed no effect on fever, but a significant reduction in oral mucositis [96].

### 4.7. Use of BC in Combination with Other Nutraceuticals

Several of the studies referred to above have used BC in combination with egg derivatives such as egg powder, but other combinations (such as proteins, carbohydrates, vitamins, probiotics, and plant polyphenols) have been evaluated [17]. Zinc carnosine, in combination with BC, reduced exercise-induced increases in intestinal permeability [97,98]. Probiotic combinations may be logical, as colostrum has prebiotic properties [37], which may help create a favorable niche for some bacteria in the microbiota.

### 4.8. Other Conditions

The place of BC as therapy for other conditions such as reflux esophagitis, peptic ulceration, or functional gastrointestinal disorders is harder to evaluate as the evidence is limited. There are preclinical studies which show that BC can reduce NSAID-induced gastric injury [37], and there is evidence that it can inhibit binding of *Helicobacter pylori* to target cells [99]. Functional dyspepsia and irritable bowel syndrome are disorders of neurosensation in the gut which have always been characterized by the absence of structural abnormalities. However, recent evidence of minor changes in permeability suggest that this is no longer absolutely true. Trials of BC in these disorders may now be informative in terms of therapy and of understanding pathophysiology.

## 5. Conclusions

Bovine colostrum is a complex biological fluid replete with growth factors, nutrients, hormones, and paracrine factors which have a range of properties likely to contribute to mucosal healing in a wide range of infective, inflammatory, and injury conditions. Evidence is building that these properties may be employed in several disorders to promote recovery or in prevention.

## Figures and Tables

**Figure 1 nutrients-13-01956-f001:**
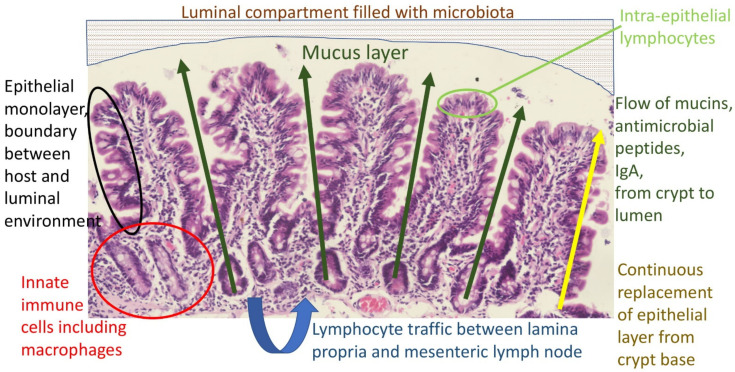
Major components of the mucosal barrier. The luminal stream is separated from the epithelium by the mucus layer, which is the physical substrate for a concentration gradient of antimicrobial molecules which acts as a repellent, protecting the stem cell compartment from microbial damage. The epithelium itself is the boundary between host and luminal environment, replaced entirely every 3–5 days and therefore dependent on proliferative drivers and repair factors. Organisms or their component molecules which traverse the epithelial boundary are then the target of innate and anamnestic immune responses.

**Table 1 nutrients-13-01956-t001:** Summary of studies of potential therapeutic uses of bovine colostrum.

First Author	Country	Study Population	Year	Number of Participants (Human Studies Only)	Conclusion
NSAID-Induced Gut Injury					
Playford et al. [37]	UK	Animal (Mice)	1999		Colostrum preparation has major beneficial effects in preventing NSAID-induced gut injury
Kim et al. [38]	Korea	Animal (rats)	2005		BC ameliorated NSAID-induced intestinal damage and bacterial translocation, especially when combined with glutamine
Cairangzhuoma et al. [39]	Japan	Animal (mice)	2013		Late BC aids in recovery and inhibition of NSAID-induced small intestinal injury in mouse model
Playford et al. [40]	UK	RCT (cross over)	2001	Acute effect, 7; long-term, 22	Compared to indomethacin alone, BC reduced the impact of indomethacin on gut permeability only in the short term
**Inflammatory Bowel Disease**					
Kanwar et al. [41]	New Zealand	Animal (mice)	2016		Bovine milk components attenuated the severity of DSS-induced colitis in mice with differing effectiveness against specific disease parameters
Filipescu et al. [42]	Italy	Animal (mice)	2018		Pre-treatment of mice with BC reduces TNBS-induced intestinal damage
Spalinger et al. [43]	Switzerland	Animal (mice)	2019		Hyperimmune BC reduces intestinal inflammation by increasing Treg cell induction while decreasing accumulation of pathogenic T cells
Playford et al. [44]	UK	Animal (mice)	2020		Combination of BC and egg synergistically reduced indomethacin and DSS-induced gut damage
Menchetti et al. [45]	Italy	Animal (mice)	2020		Pre-treatment with BC modulates the expression of genes and the count of microbes involved in the etiopathogenesis of colitis
Khan et al. [46]	UK	RCT	2002	14	Mesalazine and BC enema improved symptoms in patients with left sided colitis compared to mesalazine with placebo
**Infectious Diarrhea**					
Choudhry et al. [47]	UK	Cell culture model of microbial translocation	2020		BC reduced enteropathogen-mediated damage in Caco-2 cells
Li et al. [48]	China	Meta-analysis	2019		BC products were effective in controlling clinical symptoms and pathogenic agents in children with infectious diarrhea
Barakat et al. [49]	Egypt	RCT	2020	160	BC is effective in the treatment of acute diarrhea and can be considered as adjuvant therapy in both viral and bacterial diarrhea to prevent diarrhea-related complications
Gaensbauer et al. [50]	Guatemala	RCT	2017	301	A BC and hen’s egg derived feed reduced acute non-bloody diarrheal duration in children in a subgroup with an identified pathogen; the primary outcome was not met
Saad et al. [51]		Open multicentric, noncomparative	2016	160	BC was effective in reducing the number of episodes of URTI and diarrhea in children
Otto et al. [52]	Poland	RCT	2011	90	Hyperimmune BC is effective in protecting adult volunteers against diarrhea caused by ETEC
Kaducu et al. [53]	Uganda	RCT	2011	87	Addition of BC-based supplement is effective in treatment of HIV-associated diarrhea in adults
Eslamian et al. [54]		RCT	2019	70	BC supplementation may have beneficial effects on intestinal permeability and gastrointestinal complications in ICU-hospitalized patients
Bierut et al. [55]	Malawi	RCT	2021	267	Addition of BC and egg to complementary feeding in Malawian infants resulted in less linear growth faltering. Episodes of diarrhea and β-diversity of the 16S configuration of fecal microbiota did not differ between the treatment group and controls
**Short Bowel Syndrome**					
Paris et al. [56]	Australia	Animal (piglets)	2004		A polymeric infant formula supplemented with BC given to pig model of SBS was associated with significant increase in plasma GLP-2, suggesting the role of GLP-2 in intestinal adaptation post resection
Nagy et al. [57]		Animal (piglets)	2004		In pig model of SBS, supplementation with colostrum protein concentrate resulted in normal weight gain and features of enhanced morphologic adaptation
Pereira-Fantini et al. [58]	Australia	Animal (piglets)	2008		Following bowel resection, colostrum protein concentrate significantly increased circulating levels of IGF-1 and IGFBPs
Aunsholt et al. [59]	Denmark	Animal (piglets)	2018		Parenteral nutrition (PN) with minimal enteral nutrition with BC or formula induced similar intestinal adaption after resection
Aunsholt et al. [60]	Denmark	RCT (crossover)	2012	9	Inclusion of bovine colostrum to the diet did not improve intestinal function
Lund et al. [61]	Denmark	RCT	2012	12	BC did not significantly improve intestinal absorption, body composition, or functional tests compared with the control
**Necrotizing Enterocolitis**					
Jensen et al. [62]	Denmark	Animal (piglets)	2013		BC and human milk are both superior to formula in stimulating gut structure, function, and NEC resistance in preterm piglets
Li et al. [63]	Denmark	Animal (piglets)	2014		The maturational and protective effects on the immature intestine decreased in the order BC > mature bovine milk >whole milk powder, but all three were markedly better than formula
Støy et al. [64]		Animal (piglets)	2014		Bovine colostrum restores intestinal function after initial formula-induced inflammation in preterm pigs
Seigel et al. [65]	USA	Retrospective cohort	2013	369	Initiating oropharyngeal COL in ELBW infants in the first 2 postnatal days appears feasible and safe and may be nutritionally beneficial
Balachandran et al. [66]	India	RCT	2017	86	The use of prophylactic enteral BC in VLBW infants showed a trend toward increased stool IL-6 and features of NEC; there were no clinical benefits
Nasuf et al. [67]		Systematic review	2018		Limited available evidence currently suggests that oropharyngeal administration of mother’s colostrum starting within the first 48 h of life does not reduce the risk of NEC, late-onset infection, or death until discharge in preterm infants, including very preterm, VLBW infants
ELFIN trial investigators group [68]		RCT	2019	2203	Enteral supplementation with bovine lactoferrin (derived from milk) did not reduce the risk of late-onset infection in very preterm infants
Tao et al. [69]		Meta-analysis of RCTs	2020		BC does not reduce the incidences of NEC, late onset sepsis, and death in preterm infants, but there is a trend toward a positive effect
Sharma et al. [70]	India	RCT	2020	117	There was no significant reduction in the incidence of NEC in the BC group but there was significant reduction of 7 days of hospital stay in the BC group
Sadeghirad et al. [71]		Meta-analysis	2018		Bovine or human colostrum has no effect on severe NEC, mortality, culture-proven sepsis, feed intolerance, or length of stay

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
