# Peer review of "Colostrum Therapy for Human Gastrointestinal Health and Disease"

_nutrients, 2021, doi:10.3390/nu13061956_

Round 1

Reviewer 1 Report

The review by Chandawe and Kelly describes the beneficial effects of  bovine colostrum therapy  in mucosal healing in a wide range of gastrointestinal diseases by restoring the intestinal barrier

 The authors survey selected diseases in which the intestinal barrier is compromised and analyze studies on the efficacy of bovine colostrum in management.

 The review is of interest and well written even if it’s not specified the search methodology of the review ( which were the keywords? by the use of which databases?  years interval?

Page 3 section 4 Clinical applications: colostrum as therapy: This section could benefit from a table that can resume the studies for each pathology.

Page 6 line 251-252: Please reformulate the concept.

Where are references 30 and 31

Author Response

Responses to reviewers

Thank you for the comments of the reviewers. We have responded to their comments in bold below, and changes in the manuscript are highlighted.

Reviewer  1

The review by Chandwe and Kelly describes the beneficial effects of  bovine colostrum therapy  in mucosal healing in a wide range of gastrointestinal diseases by restoring the intestinal barrier. The authors survey selected diseases in which the intestinal barrier is compromised and analyze studies on the efficacy of bovine colostrum in management.

1 The review is of interest and well written even if it’s not specified the search methodology of the review ( which were the keywords? by the use of which databases?  years interval?

Thank you. We have added details of the search strategy (page 2).

2 Page 3 section 4 Clinical applications: colostrum as therapy: This section could benefit from a table that can resume the studies for each pathology.

We have added a table as suggested (now Table 1).

3 Page 6 line 251-252: Please reformulate the concept.

We have amended the text to make it clearer.

4 Where are references 30 and 31?

We explained at the time of submission that references 30 and  31 would be added later to permit cross-referencing with other articles in this supplement. We have now removed them.

Reviewer 2 Report

  1. Part 2 on mucosal barrier should include the terms and concepts of defensins, tight junctions between epithelial cells and the role of the endogenous microbiota

  1. Part 4 on bovine colostrum should not only quote reference 36 (which says that the present paper will be published in the Journal). It should at least briefly express why authors imagined that BC may have some therapeutic interest in humans (which requires minimal information on the composition of the product and in which forms it is provided to the volunteers of the clinical studies described below).

  1. Evidence which can be drawn from clinical studies depend on the methods used (randomized controlled trials, blinded or not) and also whether the primary endpoint was or not reached. For several papers quoted only secondary endpoints were reached with BC.

  1. Line 133: how many volunteers?

  1. Line 147: antibiotics and probiotics could be withdrawn from the list as the evidence for their efficacy is very limited (see guidelines)

  1. Line 152: 14 patients were studied

  1. Line 155 this nutrition (Modulen IBD) is rich in TGF but no TGF is added to it

  1. Reference 64: express that the positive results were not the primary endpoints of the study

  1. Line 196: can you provide information on safety?

  1. Lines 230- 232: the following sentence is not clear and should be rephrased: “However, results of clinical trials have so far not shown similar results in terms of improvements in intestinal function in children and adults, perhaps because measurement of endpoints is more.”

  1. Lines 259-260. In the absence of references supporting this sentence, I suggest to suppress it. Alternatively authors may make the efforts to express which bacteria or groups of bacteria are increased in humans and if the effect is reproducible.

  1. Lines 267-271. Of course functional GI disorders need to be better understood and treated but authors should avoid a too strong suggestion that BC might be properly tested as other treatments (statistical power, number of subjects …) and might prove effective.

  1. References 8 , 12, 44, 45 should be replaced by more recent reviews

  1. References 30, 31, 71 are lacking

Author Response

Responses to reviewers

Thank you for the comments of the reviewers. We have responded to their comments in bold below, and changes in the manuscript are highlighted.

Reviewer 2

  1. Part 2 on mucosal barrier should include the terms and concepts of defensins, tight junctions between epithelial cells and the role of the endogenous microbiota

We have added further explanation (page 2).

  1. Part 4 on bovine colostrum should not only quote reference 36 (which says that the present paper will be published in the Journal). It should at least briefly express why authors imagined that BC may have some therapeutic interest in humans (which requires minimal information on the composition of the product and in which forms it is provided to the volunteers of the clinical studies described below).

We have expanded the rationale for use of BC in humans (page 4), drawing on direct evidence that BC can enhance epithelial restitution (now reference 31).

  1. Evidence which can be drawn from clinical studies depend on the methods used (randomized controlled trials, blinded or not) and also whether the primary endpoint was or not reached. For several papers quoted only secondary endpoints were reached with BC.

We have added this important point in the new Table suggested by reviewer 1.

  1. Line 133: how many volunteers?

This information has been provided in the new Table 1

  1. Line 147: antibiotics and probiotics could be withdrawn from the list as the evidence for their efficacy is very limited (see guidelines)

We have removed probiotics as suggested (page 4), but retained antibiotics as their use is still indicated in specific conditions (eg perianal fistulation).

  1. Line 152: 14 patients were studied

Correct. This has been noted in the Table.

  1. Line 155 this nutrition (Modulen IBD) is rich in TGFb but no TGFb is added to it.

We have modified this sentence to make this clearer (page 4).

  1. Reference 64: express that the positive results were not the primary endpoints of the study

Correct. This has been made clear (page 5).

  1. Line 196: can you provide information on safety?

We have revised and expanded our previous comment on safety (page 5).

  1. Lines 230- 232: the following sentence is not clear and should be rephrased: “However, results of clinical trials have so far not shown similar results in terms of improvements in intestinal function in children and adults, perhaps because measurement of endpoints is more.”

Thank you for spotting this. Somehow this sentence got truncated. We have amended it (page 6).

  1. Lines 259-260. In the absence of references supporting this sentence, I suggest to suppress it. Alternatively authors may make the efforts to express which bacteria or groups of bacteria are increased in humans and if the effect is reproducible.

The reviewer is correct. The point of this sentence is to explain that colostrum has prebiotic effects. There is modest evidence that colostrum has prebiotic properties, for example reference 99 in the revised manuscript.

  1. Lines 267-271. Of course functional GI disorders need to be better understood and treated but authors should avoid a too strong suggestion that BC might be properly tested as other treatments (statistical power, number of subjects …) and might prove effective.

On re-reading this sentence, we feel that we are merely suggesting that changes in understanding of the pathophysiology of IBS may provide a justification for trying out BC in this condition. While IBS was always regarded as a disorder without pathology, this may not quite be true as changes in permeability have been reported. We have re-phrased this to make it clear that we do not endorse BC  for this purpose, but that we are recommending formal studies in the light of evidence that BC can ameliorate changes in permeability (page 7).

  1. References 8 , 12, 44, 45 should be replaced by more recent reviews

We have done this.

  1. References 30, 31, 71 are lacking

These have been removed and the references re-numbered.